# Lower Concentrations of Amphotericin B Combined with *Ent*-Hardwickiic Acid Are Effective against *Candida* Strains

**DOI:** 10.3390/antibiotics12030509

**Published:** 2023-03-03

**Authors:** Maria V. Sousa Teixeira, Jennyfer A. Aldana-Mejía, Márcia E. da Silva Ferreira, Niege A. J. Cardoso Furtado

**Affiliations:** Department of Pharmaceutical Sciences, School of Pharmaceutical Sciences of Ribeirão Preto, University of São Paulo, Av. do Café, s/n, Ribeirão Preto 14040-903, Brazil

**Keywords:** *ent*-hardwickiic acid, amphotericin B, antifungal combination, *Candida albicans*, *Candida krusei*, checkerboard, time–kill curves, resistant strain

## Abstract

Life-threatening *Candida* infections have increased with the COVID-19 pandemic, and the already limited arsenal of antifungal drugs has become even more restricted due to its side effects associated with complications after SARS-CoV-2 infection. Drug combination strategies have the potential to reduce the risk of side effects without loss of therapeutic efficacy. The aim of this study was to evaluate the combination of *ent*-hardwickiic acid with low concentrations of amphotericin B against *Candida* strains. The minimum inhibitory concentration (MIC) values were determined for amphotericin B and *ent*-hardwickiic acid as isolated compounds and for 77 combinations of amphotericin B and *ent*-hardwickiic acid concentrations that were assessed by using the checkerboard microdilution method. Time–kill assays were performed in order to assess the fungistatic or fungicidal nature of the different combinations. The strategy of combining both compounds markedly reduced the MIC values from 16 µg/mL to 1 µg/mL of amphotericin B and from 12.5 µg/mL to 6.25 µg/mL of *ent*-hardwickiic acid, from isolated to combined, against *C. albicans* resistant to azoles. The combination of 1 µg/mL of amphotericin B with 6.25 µg/mL of *ent*-hardwickiic acid killed all the cells of the same strain within four hours of incubation.

## 1. Introduction

Invasive candidiasis is the cause of unacceptable high mortality rates ranging from 30 to 70% in different parts of the world [1,2,3], and the treatment of life-threatening *Candida* infections has been limited to just three drug classes [2,4]. 

The emergence of *Candida* resistance to the available antifungal drugs has compromised the clinical management of this disease [1,4,5], and failure of antifungal treatment is due to multifactorial events involving molecular modifications related to drug mechanism of action and over-expression of efflux pumps, among other factors [6,7].

One mechanism of resistance of *Candida* species to azoles, for example, is the occurrence of point mutations in the *ERG11* gene [7,8]. Azoles interfere with the ergosterol biosynthesis pathway in fungal membranes by inhibiting the cytochrome P450-dependent enzyme 14α-demethylase, which is synthesized by the *ERG11* gene [7]. Mutations that resulted in amino acid substitutions decreased azoles susceptibility [7,8].

Therapeutic failures with echinocandins are also reported for *Candida* infections [6,9]. *Candida* strains with reduced susceptibility to echinocandins showed mutations in the *FKS* genes that correlated with amino acid substitutions in the 1,3-β-D-glucan synthase, the target of echinocandins [6,9].

Resistance of *Candida* species to polyenes is still uncommon compared to resistance to other antifungal drugs [6,10]. However, *Candida* species resistant to amphotericin B have been reported for clinical isolates [6,10]. A different ergosterol structure that prevents binding to the polyenes caused by several mutations has been associated with resistance to amphotericin B [6,10]. *C. albicans* resistance, for example, is associated with a substitution in *ERG11* and loss of function of *ERG5* genes (C-22 sterol desaturase) [10,11]. Isolates of other *Candida* species were reported as resistant to amphotericin B due to the inactivation of *ERG6* (C-24 sterol methyl-transferase) and *ERG2* (C-8 sterol isomerase) genes [10,12].

Although *Candida albicans* has been reported as a predominant species involved in invasive candidiasis around the world [3,4], the proportion of this infection caused by non-*albicans* species has grown in recent decades [1,3]. 

*Candida krusei* is among the non-*albicans* species whose occurrence increased during the COVID-19 pandemic when compared to pre-pandemic years [13]. This species has been reported as resistant to fluconazole [14,15] and quickly developed resistance to other antifungal drugs [14,16]. 

With the SARS-CoV-2 pandemic, there was an increase in the mortality rate due to invasive candidiasis [17,18], which oscillated between 11 and 100% according to a literature search performed in PubMed, Embase, Cochrane Library and LILACS without language restrictions, between January 2020 to February 2021 [19]. 

Data published by other authors confirmed that candidemia associated with COVID-19 also increased all-cause mortality twofold compared to patients with candidemia without COVID-19 [20].

Patients hospitalized with COVID-19 receive immunosuppressive medication that potentially increases the susceptibility of these patients to co-infections, including polymicrobial *Candida* infections [21,22]. However, according to the literature, this is not the only reason for the increase in invasive fungal infections [23]. Defective antifungal immunity in patients of COVID-19 due to a dysregulation of the immune system has been observed through the expression of exhaustion markers of natural killer cells and T cells [23]. In addition, patients with COVID-19 also have reduced fungicidal activity of neutrophils [23]. 

Amphotericin B has been recommended for the treatment of pulmonary candidiasis associated with COVID-19 infection [24], but its side effects associated with several complications after SARS-CoV-2 infection, such as kidney injury, dyspnea and hypoxia, make its use unfeasible [25,26]. 

The other classes of therapeutically available antifungal drugs are not effective alone to treat fungal co-infections in COVID-19 patients managed in the intensive care unit with prolonged immunomodulatory treatments [27,28] and have caused important side effects [29,30]. Triazoles cause hepatotoxicity, drug–drug interactions, QTc prolongation (the heart muscle takes a comparatively longer time to contract and relax than usual), skeletal fluorosis, pseudohyperaldosteronism, adrenal insufficiency, hyponatremia and hypogonadism [29,30]. The most common complications of echinocandins are thrombophlebitis, hepatotoxicity, derangement of serum transaminases, hypotension and fever, but anemia, leukopenia and thrombocytopenia have also been reported [31,32,33,34]. It should be highlighted that dexamethasone, an important drug in the treatment of COVID-19 infections, is among other drugs that interact with caspofungin [33]. Moreover, echinocandins show embryotoxicity and may not be used during pregnancy [31,34]. 

Knowing that the current antifungal drugs have numerous limitations, there is an urgent need for the discovery of antifungal agents to improve the clinical outcome of fungal infections [35,36]. 

Natural products provide innovative structural patterns with novel mechanisms of action [37,38] that can be optimized to improve efficacy and reduce toxicity. 

Among natural products, diterpenes have been recognized for their remarkable biological activities, including antifungal properties [39,40,41]. 

The clerodane-type diterpene *ent*-hardwickiic acid (Figure 1) is the major constituent of *Copaifera pubiflora* oleoresin [42] extracted from tree trunks. This tree is one of the species of the *Copaifera* genus found in Brazil, Colombia, Guyana and Venezuela [43]. 

*Copaifera* oleoresins are traditionally used by people from the Brazilian Amazonian region as an anti-inflammatory [43], antimicrobial [44] and antiparasitic [45], and literature data support the ethnopharmacological uses of this crude material [43].

The diterpene *ent*-hardwickiic acid has been highlighted as a lead compound in the search for bioactive compounds [46,47] and has been reported due to its anti-inflammatory [43], antibacterial [44], antifungal [47] and schistosomicidal activities [48]. 

Despite having several biological activities, it should be pointed out that this diterpene did not show cytotoxic activity against normal and cancer human cell lines [49]. 

This study reports for the first time the in vitro interaction between *ent*-hardwickiic acid and amphotericin B by using the checkerboard microdilution method against *C. albicans* and *C. krusei* strains, including a *C. albicans* strain resistant to azoles isolated from bloodstream infections in a tertiary care hospital in Brazil [50]. In addition, time–kill assays were performed to assess the fungistatic or fungicidal nature of different combinations of *ent*-hardwickiic acid and amphotericin B concentrations. 

Considering that only one new azole and two new formulations of posaconazole have been launched in the market in the last decade [51] and that the need for new antifungal drugs is urgent, the strategy of combining *ent*-hardwickiic acid with amphotericin B was shown to be potentially effective at a low concentration of amphotericin B against *Candida* strains. 

## 2. Results

The minimum inhibitory concentrations (MICs) of amphotericin B and *ent*-hardwickiic acid were first determined using the broth microdilution method. Amphotericin B showed MIC values of 8 µg/mL against *C. albicans* ATCC 10231 and *C. krusei* ATCC 6258 and 16 µg/mL against a *C. albicans* strain resistant to azoles. The MIC values of *ent*-hardwickiic acid were smaller than those found for amphotericin B (6.25 µg/mL against *C. albicans* ATCC 10231, 3.12 µg/mL against *C. krusei* ATCC 6258 and 12.5 µg/mL against a *C. albicans* strain resistant to azoles). 

MIC values of fluconazole were also determined for *C. albicans* ATCC 10231 (12.5 µg/mL) and the quality control strain *Candida parapsilosis* ATCC 22019 (4 µg/mL) to assure that the antifungal microdilution test was performed appropriately [52,53]. Our results were reproducible, and the MIC values are within the proposed range for these strains.

It should be pointed out that *C. krusei* is considered intrinsically resistant to fluconazole [53], and the clinical isolate of *C. albicans* used in this study also showed resistance to fluconazole [50]. In this study, the MIC values of fluconazole against *C. krusei* ATCC 6258 and *C. albicans* resistant strain were 25 ug/mL and greater than 100 ug/mL, respectively.

The combination of amphotericin B (1) and *ent*-hardwickiic acid (2) was then assessed by using the checkerboard microdilution method and synergistic (ƩFIC ≤ 0.5) and additive (ƩFIC > 0.5) interactions of compounds 1 and 2 were found for tested strains (Table 1). Antagonism was not detected. 

The combination of both compounds at determined concentrations markedly reduced the MIC values, and a synergistic effect was detected when 4 µg/mL of amphotericin B was combined with 3.12 µg/mL of *ent*-hardwickiic acid against a *C. albicans* strain resistant to azoles. Additive effects were detected with 1 and 2 µg/mL of amphotericin B combined with 6.25 µg/mL of *ent*-hardwickiic acid, with 0.125 µg/mL of amphotericin B combined with 1.00 µg/mL of *ent*-hardwickiic acid and with 8 µg/mL of amphotericin B combined with 1.56 µg/mL of *ent*-hardwickiic acid against the same resistant strain. 

Synergistic effects were detected against the reference strains of *C. albicans* and *C. krusei* in the range of amphotericin B concentrations from 0.031 µg/mL to 2 µg/mL and from 0.0156 µg/mL to 2 µg/mL, respectively. The *ent*-hardwickiic acid concentrations varied in the same assay from 0.39 µg/mL to 3.12 µg/mL and from 0.195 µg/mL to 0.78 µg/mL, respectively. 

In order to assess the fungistatic or fungicidal nature of different combinations of *ent*-hardwickiic acid and amphotericin B concentrations, time–kill assays were performed using four combinations of amphotericin B and *ent*-hardwickiic acid concentrations for each strain that resulted in growth inhibition at amphotericin B concentrations lower than the MIC value of this antifungal agent alone. The four selected combinations for each strain are presented in Table 2. 

The fourth combination containing 4 µg/mL of amphotericin B and 3.12 µg/mL of *ent*-hardwickiic acid killed all *C. albicans* resistant strain cells within 2 h (Figure 2a). The combination of 1 µg/mL of amphotericin B with 6.25 µg/mL of *ent*-hardwickiic acid killed all the cells of the same strain within 4 h of incubation. The other antifungal combinations did not show fungicidal activity within the 24 h incubation period, but exhibited a significant reduction in the growth of this strain.

Among the tested combinations of *ent*-hardwickiic acid and amphotericin B concentrations against the reference strain of the *C. albicans*, the combination of 0.5 µg/mL of amphotericin B and 0.78 µg/mL of *ent*-hardwickiic acid was the one that showed the highest growth reduction within 24 h (Figure 2b). During this period of 24 h, *ent*-hardwickiic acid and amphotericin B combinations exhibited a significant reduction in the growth of *C. albicans* reference strain, but the fungicidal point was not detected.

The same behavior was observed for the time–kill curves of *ent*-hardwickiic acid and amphotericin B combinations against the *C. krusei* reference strain. All the curves showed a significant reduction in growth during 24 h without achieving the fungicidal point in this period. The combination of 0.25 µg/mL of amphotericin B and 0.39 µg/mL of *ent*-hardwickiic acid showed the highest growth reduction within 24 h (Figure 2c). 

Considering the results obtained from time–kill assays, it should be highlighted that the fungicidal activity of two combinations of *ent*-hardwickiic acid and amphotericin B concentrations against *C. albicans* resistant strain was very fast (2 to 4 h), which can be clinically relevant.

## 3. Discussion

The aim of this study was to evaluate for the first time the potential of *ent*-hardwickiic acid combined with amphotericin B against *Candida* strains. 

Amphotericin B was licensed in 1959 and after more than sixty years is still the main antifungal agent used to treat invasive fungal infections [54,55]. However, its principal chronic adverse effect is nephrotoxicity, whose clinical manifestations range from hypokalemia to kidney insufficiency [56].

Among several complications that might arise after SARS-CoV-2 infection is the acute kidney injury that affects over a quarter of patients hospitalized with COVID-19 disease [57]. The clinical management of these patients includes hemodynamic support and avoidance of nephrotoxic drugs [58]. 

Drug combination strategies have the potential to reduce the risk of side effects due to a reduction of effective dose of each compound without loss of therapeutic efficacy [59].

In this study, the combination of *ent*-hardwickiic acid and amphotericin B markedly reduced the MIC values when compared with those of drugs alone. The combination was effective in using lower concentrations of each compound than those needed to achieve the same effect of each isolated compound. In addition, two combinations of *ent*-hardwickiic acid and amphotericin B concentrations exhibited fungicidal activity against *C. albicans* resistant strain after 2–4 h of incubation. 

The combination of 1 µg/mL of amphotericin B with 6.25 µg/mL of *ent*-hardwickiic acid killed all the cells of *C. albicans* resistant strain within four hours of incubation. This concentration of amphotericin B in plasma has not been associated with toxic effects and drug discontinuation [60]. According to the literature, the pharmacodynamic characteristics of amphotericin B indicate that after the administration of doses of 0.6 to 3.0 mg/kg of body weight/day of amphotericin B deoxycholate (Bristol-Myers Squibb), the mean maximum concentrations (C_max_s) achieved in serum are 1.1 to 3.6 µg/mL [60,61]. In the presence of serum, amphotericin B loses its fungicidal activity, but remains with its fungistatic activity [62].

There are no studies yet about the stability of *ent*-hardwickiic acid in human serum, but this compound has been highlighted as a lead compound in the search for bioactive compounds [46,47]. In previous studies, this diterpene showed fungistatic and fungicidal effects against *C. glabrata* at lower concentrations than fluconazole and its derivatives obtained by biotransformation reactions exhibited potent antifungal activity [47].

Regarding the toxicity of *ent*-hardwickiic acid, a study carried out in normal and tumor human cell lines showed that this diterpene was not cytotoxic to the tested cell lines [43,49,63], as well as to the RAW 264.7 cells, which are monocyte/macrophage-like cells reported as an appropriate model of macrophages [45]. In addition, this compound did not affect the animal’s locomotor capacity in open-field and rotarod tests [43]. 

Many marketed drugs have a natural product origin, and the majority of these successful natural products were formulated to interact with biological systems to achieve their therapeutic potential [64,65]. 

Natural products may also provide different mechanisms of action, since these compounds are optimized by evolution to be useful in the defense of organisms [66]. As an example to be cited, macrocyclic diterpenes were able to overcome multidrug resistance in *C. albicans* as potent inhibitors of drug efflux pumps [67].

With regard to *ent*-hardwickiic acid, there is only one study reporting the mechanism of action of this diterpene against *Streptococcus mutans* (ATCC 25175) and *Porphyromonas gingivalis* (ATCC 33277) [68]. The authors performed assays to determine cell membrane integrity by leakage through the bacterial membrane of nucleic acids and protein. The results indicated that the diterpene *ent*-hardwickiic acid damaged the *S. mutans* and *P. gingivalis* cell membranes, causing cellular component release followed by the release of cytoplasmic material [68]. Further studies are necessary to elucidate the mechanism of the antifungal action of this compound. 

The interest in natural products to provide drug leads has been revitalized mainly with the aim of overcoming the resistance of microorganisms to antimicrobial agents [66].

In conclusion, the results of the present study indicate that the combination of amphotericin B and the natural product *ent*-hardwickiic acid has the potential to inspire the development of treatment options for life-threatening *Candida* infections. 

## 4. Materials and Methods

### 4.1. Candida Strains 

*Candida albicans* ATCC 10231, *C. krusei* ATCC 6258 and *C. parapsilosis* ATCC 22019 were acquired from American Type Culture Collection (ATCC, Rockville, MD, USA). The *C. albicans* strain resistant to azoles was isolated from bloodstream infections in a tertiary care hospital in Brazil using the Bactec™ 9240 system (Becton & Dickinson, Franklin Lanes, NJ, USA) and provided for this study by Prof. Dr. Márcia E. da Silva Ferreira. This strain was identified with the VITEK^®^ 2 system (BioMérieux, Marcy l’Étoile, France) and by using molecular techniques [50].

### 4.2. Antifungal Agents

Amphotericin B and fluconazole were acquired from Sigma-Aldrich, and *ent*-hardwickiic acid was isolated from *Copaifera pubiflora* oleoresin according to Teixeira and co-workers [47].

The authorizations to undertake scientific studies with *C. pubiflora* oleoresin were issued under the numbers 35143-1 and 010225/2014-5 from the Brazilian Council for Authorization and Information on Biodiversity (SIBIO/ICMBio/MMA/BRASIL) and Genetic Heritage Management (CGEN/MMA/BRASIL), respectively.

### 4.3. Minimum Inhibitory Concentration of Antifungal Compounds

The minimum inhibitory concentration values of amphotericin B and *ent*-hardwickiic acid against *Candida* strains were first determined in triplicate by using the broth microdilution method in 96-well microplates according to the recommendations of the Clinical and Laboratory Standards Institute (document M27-A4) [69]. Amphotericin B and *ent*-hardwickiic acid were dissolved in dimethyl sulfoxide (Merck, Saint Louis, USA) and diluted in RPMI 1640 medium to achieve concentrations ranging from 16 µg/mL to 0.0156 µg/mL and from 100 µg/mL to 0.19 µg/mL, respectively. The final content of DMSO was 5% (v/v), and this solution was used as negative control. The fungal inoculum was adjusted to yield a cell concentration of 2.5 × 10^3^ CFU/mL. The following controls were included: one inoculated and one non-inoculated well to verify the adequacy of the broth for organism growth and the medium sterility, respectively. Fluconazole was used as positive control and its MIC value was also determined for the quality control strain *C. parapsilosis* ATCC 22019 to assure that the antifungal microdilution test was performed appropriately [52,53]. The 96-well microplates were incubated at 35 °C for 24 h. After the incubation period, the microorganism viability was also measured by adding 30 µL of resazurin solution (0.02%) to the microplates to confirm the MIC values determined visually [70]. 

### 4.4. Checkerboard Microdilution Method

The in vitro interactions between amphotericin B and *ent*-hardwickiic acid were investigated by using the checkerboard microdilution method in 96-well microplates as previously described with adaptations [71]. Amphotericin B and *ent*-hardwickiic acid were dissolved in dimethyl sulfoxide, and stock solutions of both compounds were prepared in RPMI 1640 medium in the range of concentrations from 4- to 8-fold more concentrated than the highest concentration of each compound to be tested.

In each well of the microplate, 100 µL of growth medium was added, and serial twofold dilutions of amphotericin B and *ent*-hardwickiic acid stock solutions were mixed in each well, resulting in 77 combinations (Appendix A).

The MIC values of the isolated compounds were again determined by inoculating only amphotericin B and *ent*-hardwickiic acid in row H (12-2) and column 1 (A–G), respectively. One well without the antifungals was added as growth control. 

The final inoculum was adjusted to yield a cell concentration of 2.5 × 10^3^ CFU/mL. A mirror plate without microorganisms and with the same concentrations of compounds was prepared to be used as optical density background in a microplate reader. Both microplates were incubated at 35 °C for 24 h.

The growth in each well was quantified spectrophotometrically at 530 nm in a microplate reader, and the MIC values for each combination of compounds were defined as the concentration of compounds combination or the concentration of isolated compound that reduces microbial growth by more than 80% [71]. 

The interactions between amphotericin B (1) and *ent*-hardwickiic acid (2) in different combinations of concentrations were determined based on the calculated coefficient of the sum of fractional inhibitory concentration (ƩFIC) [72]. The ƩFIC is calculated according to the formula:ƩFIC = FIC_1_ + FIC_2_,(1)
where
FIC_1_ = MIC_1 in combination_/MIC_1_,(2)
and
FIC_2_ = MIC_2 in combination_/MIC_2_(3)

The results can be interpreted as follow: ƩFIC ≤ 0.5: synergistic, ƩFIC > 0.5 to ≤ 1: additive, ƩFIC > 1 to ≤ 4: indifferent and ƩFIC > 4: antagonistic. 

### 4.5. Time–Kill Assays 

Time–kill assays were performed in triplicate for four combinations of amphotericin B and *ent*-hardwickiic acid concentrations following the procedures proposed for the time–kill evaluation of antibacterial agents [73] with adaptations. The assays were also carried out with microorganisms without antifungal agents. 

The final inoculum was adjusted to yield a cell concentration of 2.5 × 10^3^ CFU/mL.

Microplates containing the combinations of compounds and the microorganisms were incubated at 35 °C for 24 h. During this period, aliquots (20 µL) of each well were removed, diluted when necessary and spread onto Sabouraud dextrose agar for counting of viable colonies at predetermined time points (0, 2, 4, 6, 12 and 24 h). The lower limit of accurate and reproducible detectable colony counts was 100 CFU/mL. 

Time–kill curves were built by plotting log_10_ CFU/mL versus time with the aid of the Prism software (version 5.0; GraphPad Software, Inc., Boston, MA, USA).

## Figures and Tables

**Figure 1 antibiotics-12-00509-f001:**
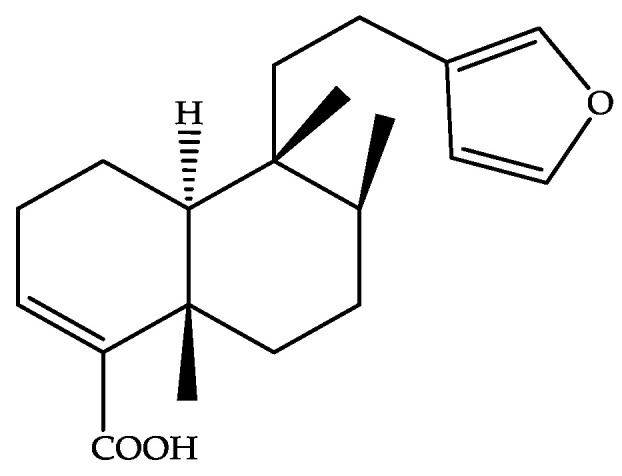
Chemical structure of *ent*-hardwickiic acid.

**Figure 2 antibiotics-12-00509-f002:**
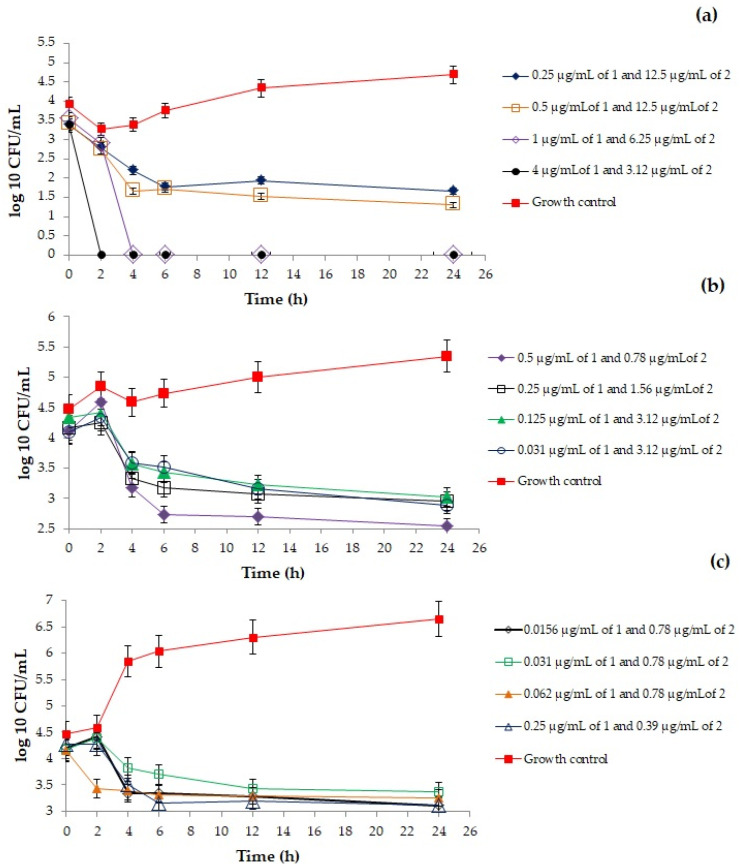
Time–kill curves of amphotericin B (1) and *ent*-hardwickiic acid (2) combinations against *Candida* strains: (**a**) *C. albicans* resistant strain; (**b**) *C. albicans* ATCC 10231; (**c**) *C. krusei* ATCC 6258. The error bars indicate standard deviations based on three replicates.

**Table 1 antibiotics-12-00509-t001:** Main interactions of amphotericin B (1) with *ent*-hardwickiic acid (2) in vitro against *Candida* strains using checkerboard microdilution method.

	Strains											
	*C. albicans* Resistant Strain				*C. albicans* ATCC 10231				*C. krusei* ATCC 6258			
	MIC 1 Alone (µg/mL): 16		MIC 2 Alone (µg/mL): 12.5		MIC 1 Alone (µg/mL): 8		MIC 2 Alone (µg/mL): 6.25		MIC 1 Alone (µg/mL): 8		MIC 2 Alone (µg/mL): 3.12	
MIC Combinated 1 (µg/mL)	MIC Combinated 2 (µg/mL)	Ʃ FIC		Interaction Type *	MIC Combinated 2 (µg/mL)	Ʃ FIC		Interaction Type *	MIC Combinated 2 (µg/mL)	Ʃ FIC		Interaction Type *
16	1.56	1.12		Indifferent	0.39	2.06		Indifferent	0.195	2.06		Indifferent
8	1.56	0.62		Additive	0.39	1.06		Indifferent	0.195	1.06		Indifferent
4	3.12	0.50		Synergism	0.39	0.56		Additive	0.195	0.56		Additive
2	6.25	0.62		Additive	0.39	0.31		Synergism	0.195	0.31		Synergism
1	6.25	0.56		Additive	0.78	0.24		Synergism	0.39	0.25		Synergism
0.5	12.5	1.03		Indifferent	0.78	0.18		Synergism	0.39	0.18		Synergism
0.25	12.5	1.01		Indifferent	1.56	0.28		Synergism	0.39	0.15		Synergism
0.125	12.5	1.00		Additive	3.12	0.50		Synergism	0.39	0.14		Synergism
0.062	25	2.00		Indifferent	3.12	0.50		Synergism	0.78	0.25		Synergism
0.031	25	2.00		Indifferent	3.12	0.50		Synergism	0.78	0.24		Synergism
0.0156	25	2.00		Indifferent	6.25	1.00		Additive	0.78	0.25		Synergism

* Interpretations of interactions type: ƩFIC ≤ 0.5: synergistic, ƩFIC > 0.5 to *≤* 1: additive, ƩFIC *>* 1 to ≤ 4: indifferent and ƩFIC *>* 4: antagonistic.

**Table 2 antibiotics-12-00509-t002:** Selected concentrations of amphotericin B (1) and *ent*-hardwickiic acid (2) (µg/mL) based on the checkerboard assay for time–kill assays of *Candida* strains.

	Strains					
	*C. Albicans* Resistant Strain		*C. albicans* ATCC 10231		*C. krusei* ATCC 6258	
Amphotericin B (1) and *Ent*-Hardwickiic Acid (2) (µg/mL)						
Selected Combinations	1	2	1	2	1	2
1°	0.25	12.5	0.031	3.12	0.0156	0.78
2°	0.5	12.5	0.125	3.12	0.031	0.78
3°	1	6.25	0.25	1.56	0.062	0.78
4°	4	3.12	0.5	0.78	0.25	0.39

## Data Availability

Not applicable.

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
