# Peer review of "Lower Concentrations of Amphotericin B Combined with Ent-Hardwickiic Acid Are Effective against Candida Strains"

_antibiotics, 2023, doi:10.3390/antibiotics12030509_

Round 1

Reviewer 1 Report

This communication manuscript focuses on the synergistic effect between amphotericin B and ent-hardwickiic acid against Candida strains. It demonstrates clinical potential for combining these two compounds and tackle the increasing Candida infection problem, especially during the SARS-CoV-2 pandemic.

I found the article easy to understand, and the conclusion is clear. The overall data presentation is clear, the references are appropriate, and the methods are well written. There are a few minor issues that need some changes.

Line 33: "word" -> "world"

Line 49: "can provide" -> "provide"

Line 81: "MICS" -> "MICs"

Line 101, Table 1:

I would suggest to use two digits for all numbers. For example, 1.1 -> 1.10.

For the C. krusei ATCC 6258 line 3, the FICI is 0.56, so it should be "Synergism" instead of "Indifferent"?

Figure 2 (b), the legend does not correlates to the concentration mentioned in Table 2.

Author Response

Dear Reviewer,

We deeply appreciate your consideration in helping us improving our manuscript ID: antibiotics-2161977. As requested, we have verified the annotated comments and following we are detailing step by step how we have dealt with all the comments.

As requested by the editor, we highlighted all changes in the revised manuscript.

This communication manuscript focuses on the synergistic effect between amphotericin B and ent-hardwickiic acid against Candida strains. It demonstrates clinical potential for combining these two compounds and tackle the increasing Candida infection problem, especially during the SARS-CoV-2 pandemic.

I found the article easy to understand, and the conclusion is clear. The overall data presentation is clear, the references are appropriate, and the methods are well written. There are a few minor issues that need some changes.

  1. Line 33: "word" -> "world"

Thank you. We have corrected.

  1. Line 49: "can provide" -> "provide"

Thank you. We have rephrased.

  1. Line 81: "MICS" -> "MICs"

Thank you. We have rephrased.

  1. Line 101, Table 1:

I would suggest to use two digits for all numbers. For example, 1.1 -> 1.10.

Thank you. We have included two digits.

  1. For the C. krusei ATCC 6258 line 3, the FICI is 0.56, so it should be "Synergism" instead of "Indifferent"?

Thank you. We have corrected according to literature (Saracino IM, Foschi C, Pavoni M, Spigarelli R, Valerii MC, Spisni E. Antifungal Activity of Natural Compounds vs. Candida spp.: A Mixture of Cinnamaldehyde and Eugenol Shows Promising In Vitro Results. Antibiotics (Basel). 2022 Jan 8;11(1):73. doi: 10.3390/antibiotics11010073. PMID: 35052950; PMCID: PMC8773119; Sharifzadeh A, Khosravi AR, Shokri H, Tari PS. Synergistic anticandidal activity of menthol in combination with itraconazole and nystatin against clinical Candida glabrata and Candida krusei isolates. Microb Pathog. 2017 Jun;107:390-396. doi: 10.1016/j.micpath.2017.04.021. Epub 2017 Apr 19. PMID: 28431915).

  1. Figure 2 (b), the legend does not correlates to the concentration mentioned in Table 2.

Thank you. We have corrected.

We take this opportunity to thank the editor and referees for the pertinent and very important observations made to improve the quality of this manuscript.

We have taken into consideration all the issues mentioned by the editor and referees and we thank you again for taking your precious time to improve our manuscript.

We thank you in advance for your consideration and we are looking forward to hearing from you soon.

Yours sincerely,

Prof. Dr. Niege Araçari Jacometti Cardoso Furtado

Reviewer 2 Report

The manuscript “Lower concentrations of amphotericin B combined with ent-hardwickiic acid are effective against Candida strains” proposes using ent-hardwickiic acid as a synergist for amphotericin B. Some corrections and improvements are necessary to the document.

The introduction is written in short sentences, which seem to be isolated ideas; it is necessary to reorganize the introduction with structured paragraphs.

They justify the study based on secondary illnesses from COVID-19; however, it is necessary to place the statistics of patients who suffer complications from candidiasis and what is the mortality rate of patients from candidiasis and COVID-19.

Line 46, mention the limitations of current antifungal compounds.

If the positive control was fluconazole, why was it not evaluated against C. krusei ATCC6258.

Review the reference on line 93, is 34 correct?

Table 1 is unclear; if the mic is 16 μg/mL, they should have analyzed the MIC 4 dilutions up and 4 down in the checkboard.

Table 1: The FIC calculations do not match the displayed results; please verify or send an example equation

Table 2, why not evaluate the concentration of 2 μg/mL of Amphotericin B in C. albicans resistant strain?

Fig. 2, place mL, not ml

Fig. 2b, the values of ent-hardwickiic acid do not coincide with Table 2.

Place the MIC of the positive control (Fluconazole) in Table 1 and the time-kill assay. With these comparisons, the results would have greater significance in treatments against candidiasis.

Lines 237 to 239, place a citation where resazurin has been used in yeast stains.

Place in supplementary material the photographs of the Checkerboard microdilution methods, edited with those with the different concentrations.

Line 254, Because at such low concentrations of CFU/mL.

Line 261, specify how they determined that the combination of the two compounds inhibited 80% of the CFU.

Did they perform the assays in triplicate? I mean, are they reproducible?

The Time-kill tests had to be carried out in triplicate, adding the standard deviations in Figure 2.

Author Response

Dear Reviewer,

We deeply appreciate your consideration in helping us improving our manuscript ID: antibiotics-2161977. As requested, we have verified the annotated comments and following we are detailing step by step how we have dealt with all the comments.

As requested by the editor, we highlighted all changes in the revised manuscript.

The manuscript “Lower concentrations of amphotericin B combined with ent-hardwickiic acid are effective against Candida strains” proposes using ent-hardwickiic acid as a synergist for amphotericin B. Some corrections and improvements are necessary to the document.

  1. The introduction is written in short sentences, which seem to be isolated ideas; it is necessary to reorganize the introduction with structured paragraphs.

As requested, we have rewritten the introduction with structured paragraphs.

  1. They justify the study based on secondary illnesses from COVID-19; however, it is necessary to place the statistics of patients who suffer complications from candidiasis and what is the mortality rate of patients from candidiasis and COVID-19.

As requested, we have included in the text (Introduction, page 2) as follows:

With the SARS-CoV-2 pandemic, there was an increase in the mortality rate due to invasive candidiasis [17,18], which oscillated between 11 and 100% according to a literature search performed in PubMed, Embase, Cochrane Library, and LILACS without language restrictions, between January 2020 to February 2021 [19]. 

Data published by other authors confirmed that candidemia associated with COVID-19 also increased all-cause mortality two-fold compared to patients with candidemia without COVID-19 [20].”

  1. Line 46, mention the limitations of current antifungal compounds.

We have included in the text (Introduction, page 2) as follows:

Amphotericin B has been recommended for the treatment of pulmonary candidiasis associated with COVID-19 infection [24], but its side effects associated with several complications of SARS-CoV-2 infection, such as kidney injury, dyspnea, hypoxia, makes its use unfeasible [25,26].

The other classes of therapeutically available antifungal drugs are not effective alone to treat fungal co-infections in COVID-19 patients managed in intensive care unit with prolonged immunomodulatory treatments [27,28] and have caused important side effects [29,30]. Triazoles cause hepatotoxicity, drug-drug interactions, QTc prolongation (the heart muscle takes a comparatively longer time to contract and relax than usual), skeletal fluorosis, pseudohyperaldosteronism, adrenal insufficiency, hyponatraemia and hypogonadism [29,30]. The most common complications of echinocandins are thrombophlebitis, hepatotoxicity, derangement of serum transaminases, hypotension and fever, but anemia, leukopenia and thrombocytopenia have also been reported [31-34]. It should be highlighted that dexamethasone, an important drug in the treatment of COVID-19 infections, are among other drugs that interact with caspofungin [33]. Moreover, echinocandins show embryo toxicity and may not be used during pregnancy [31,34].”

  1. If the positive control was fluconazole, why was it not evaluated against C. krusei ATCC6258.

 We evaluated and determined the minimum inhibitory concentration value of   fluconazole against C. krusei ATCC 6258 that was equal to 25 μg/mL.

We did not include it in the text earlier because according to CLSI guidelines (Clinical and Laboratory Standards Institute (CLSI). Performance standards for antifungal susceptibility testing of yeasts. 1st ed. CLSI supplement M60. Wayne, USA, 2017), there is no minimum inhibitory concentration breakpoints for in vitro broth dilution susceptibility testing of C. krusei and antifungal agents.

In the text we have mentioned “It should be pointed out that C. krusei is considered intrinsically resistant to fluconazole [53] and the clinical isolate of C. albicans used in this study also showed resistance to fluconazole [50].” 

We mentioned fluconazole MIC values against C. albicans ATCC 10231 and C. parapsilosis ATCC 22019, for which there are tabulated values in the CLSI supplement M60 guideline as follows:

“MIC values of fluconazole were also determined for C. albicans ATCC 10231 (12.5 µg/ mL) and the quality control strain Candida parapsilosis ATCC 22019 (4 µg/ mL) to assure that the antifungal microdilution test was performed appropriately [52,53]. Our results were reproducible and the MIC values are within the proposed range for these strains.”

We have included in the revised version of this manuscript (page 3) the MIC value of fluconazole against C. krusei and as requested in item 11 below, the MIC value of fluconazole against C. albicans resistant strain, as follows: 

“It should be pointed out that C. krusei is considered intrinsically resistant to fluconazole [53] and the clinical isolate of C. albicans used in this study also showed resistance to fluconazole [50]. In this study, the MIC values of fluconazole against C. krusei ATCC 6258 and C. albicans resistant strain were of 25 ug/ mL and greater than 100 ug/ mL, respectively.       

  1. Review the reference on line 93, is 34 correct?

Yes, this is the reference of CLSI supplement M60 (Clinical and Laboratory Standards Institute (CLSI). Performance standards for antifungal susceptibility testing of yeasts. 1st ed. CLSI supplement M60. Wayne, USA, 2017) in which it states “isolates of C. krusei are assumed to be intrinsically resistant to fluconazole, so their MICs should not be interpreted using this scale”. In the Revised Version of this manuscript this reference is number 53.  

  1. Table 1 is unclear; if the mic is 16 μg/mL, they should have analyzed the MIC 4 dilutions up and 4 down in the checkboard.

According to the Method Article (Bellio, P.; Fagnani, L.; Nazzicone, L.; Celenza, G. New and simplified method for drug combination studies by checkerboard assay. MethodsX. 2021, 8, 101543, doi: 10.1016/j.mex.2021.101543. PMID: 34754811; PMCID: PMC8563647) amphotericin B and ent-hardwickiic acid were dissolved in dimetil sulfoxide and stock solutions of both compounds were prepared in RPMI 1640 medium in the range of concentrations from 4 to 8-fold more concentrated than the highest concentration of each compound to be tested. From these stock solutions, both compounds were combined after serial dilutions to achieve 77 combinations of concentrations. In addition to the 77 combinations of concentrations of the two compounds, each compound was also evaluated separately by this method in the same microplate and the MIC values were identical to those determined by the broth microdilution method. As requested in item 13 below, the concentrations of each compound after serial dilutions are shown in the Supplementary material (Figure S1).  

  1. Table 1: The FIC calculations do not match the displayed results; please verify or send an example equation

Thank you. We have corrected the wrong ones. The equation is in Materials and Methods section, page 9, item 4.4.

  1. Table 2, why not evaluate the concentration of 2 μg/mL of Amphotericin B in C. albicans resistant strain?

This concentration was evaluated by broth microdilution method and checkerboard microdilution method as mentioned in Material and Methods section, items 4.3 and 4.4 (page 8).  

  1. Fig. 2, place mL, not ml

Thank you. We have corrected in the whole text.

  1. Fig. 2b, the values of ent-hardwickiic acid do not coincide with Table 2.

Thank you. We have corrected.

  1. Place the MIC of the positive control (Fluconazole) in Table 1 and the time-kill assay. With these comparisons, the results would have greater significance in treatments against candidiasis.

The results shown in Table 1 are from the Checkerboard microdilution method with the combination of amphotericin B and ent-hardwickiic acid. Considering that fluconazole and other triazoles are not effective alone to treat fungal co-infections in COVID-19 patients managed in intensive care unit with prolonged immunomodulatory treatments and have caused important side effects due to drud-drug interactions, among the objectives of this study was not the establishing of combinations with fluconazole.

We have included as mentioned before in item 4 the MIC value of fluconazole against C. krusei ATCC 6258 and we also presented the MIC values of fluconazole against C. albicans ATCC 10231 and C. parapsilosis ATCC 22019. We have also included in this revised version that the MIC value of fluconazole against C. albicans resistant strain was greater than 100 µg/mL.     

  1. Lines 237 to 239, place a citation where resazurin has been used in yeast stains.

Thank you. We have replaced the reference of Palomino et al. 2002 with one of the references reporting the use of resazurin in yeasts (Raber HF, Sejfijaj J, Kissmann AK, Wittgens A, Gonzalez-Garcia M, Alba A, Vázquez AA, Morales Vicente FE, Erviti JP, Kubiczek D, Otero-González A, Rodríguez A, Ständker L, Rosenau F. Antimicrobial peptides Pom-1 and Pom-2 from Pomacea poeyana are active against Candida aurisC. parapsilosis and C. albicans biofilms. Pathogens. 2021 Apr 20;10(4):496. doi: 10.3390/pathogens10040496. PMID: 33924039; PMCID: PMC8072573). 

  1. Place in supplementary material the photographs of the Checkerboard microdilution methods, edited with those with the different concentrations.

We have included a supplementary material showing the concentrations of compounds in the checkerboard microdilution method.

  1. Line 254, Because at such low concentrations of CFU/mL.

In our study, we used the recommendations of the Clinical and Laboratory Standards Institute (document M27-A4 - Clinical and Laboratory Standards Institute (CLSI). Reference method for broth dilution antifungal susceptibility testing of yeasts. 4.ed. CLSI standard M27. Wayne, USA, 2017). In this guideline, the inoculum must be prepared to achieve 0.5 x 103 to 2.5 x 103 CFU/mL when the wells are inoculated. The same inoculum size was used for all experiments in order to compare the results of the different methods. This inoculum size has been used in other recent studies as in the manuscript published by Saracino IM, Foschi C, Pavoni M, Spigarelli R, Valerii MC, Spisni E. Antifungal Activity of Natural Compounds vs. Candida spp.: A Mixture of Cinnamaldehyde and Eugenol Shows Promising In Vitro Results. Antibiotics (Basel). 2022 Jan 8;11(1):73. doi: 10.3390/antibiotics11010073. PMID: 35052950; PMCID: PMC8773119.      

  1. Line 261, specify how they determined that the combination of the two compounds inhibited 80% of the CFU.

The growth in each well was quantified spectrophotometrically at 530 nm in a microplate reader. Two microplates were compared. One complete with compounds and microorganisms in which the well H1 is the growth control (100 % growth without antibiotics) and the other one, a mirror plate, without microorganisms and with the same reagents to obtain no-growth control. The mirror plate is used to obtain optical density (OD) background in the data analysis.

After reading, the percentage of growth in each well was calculated as:

  ODdrug combination well - ODbackground     X 100

   OD drug free well - ODbackground

  1. Did they perform the assays in triplicate? I mean, are they reproducible?

Yes, all experiments were performed in triplicate and are reproducible.

  1. The Time-kill tests had to be carried out in triplicate, adding the standard deviations in Figure 2.

The time-kill assays were performed in triplicate and the standard deviations are in figure 2.

We take this opportunity to thank the editor and referees for the pertinent and very important observations made to improve the quality of this manuscript.

We have taken into consideration all the issues mentioned by the editor and referees and we thank you again for taking your precious time to improve our manuscript.

We thank you in advance for your consideration and we are looking forward to hearing from you soon.

Yours sincerely,

Prof. Dr. Niege Araçari Jacometti Cardoso Furtado

Reviewer 3 Report

Lower concentrations of amphotericin B combined with ent-2 hardwickiic acid are effective against Candida strains

The manuscript has a very complete and actualized introduction, bolding the importance of their work. It is well written, and the bibliography is more than correct.

The potential of the compound has been shown. And it is very promising.

However, more toxicity studies must be done, not only in normal cells (the authors mention absence of toxicity on cancer cells) but in animals.

Besides, the most used antifungal drugs are azoles and the authors do not show any experiment in combination with azoles.

Nowadays, these kind of studies should be combined with others targeted to find out some clues about the mechanism of action of the new compound. Although natural products are a unvaluable source of new drugs, without more toxicity studies and explanations about their mechanism of action, these new compounds will be restricted to the lab.

Author Response

Dear Reviewer,

We deeply appreciate your consideration in helping us improving our manuscript ID: antibiotics-2161977. As requested, we have verified the annotated comments and following we are detailing step by step how we have dealt with all the comments.

As requested by the editor, we highlighted all changes in the revised manuscript.

The manuscript has a very complete and actualized introduction, bolding the importance of their work. It is well written, and the bibliography is more than correct.

The potential of the compound has been shown. And it is very promising.

  1. However, more toxicity studies must be done, not only in normal cells (the authors mention absence of toxicity on cancer cells) but in animals.

The study mentioned in the previous version of the manuscript was performed with normal and tumor human cell lines. We have included information about this study in more detail along with information available in the literature on this topic in the discussion (page 7) as follows:

“Regarding toxicity of ent-hardwickiic acid, a studied carried out in normal and tumor human cell lines showed that this diterpene was not cytotoxic to the tested cell lines [43,49,63], as well as to the RAW 264.7 cells, which are monocyte/macrophage-like cells reported as an appropriate model of macrophages [45]. In addition, this compound did not affect animal’s locomotor capacity in open-field and rotarod tests [43].”     

  1. Besides, the most used antifungal drugs are azoles and the authors do not show any experiment in combination with azoles.

We have included in the Introduction (page 2) the reason why it was not the objective of this study to perform combination studies with triazoles as follows:

“The other classes of therapeutically available antifungal drugs are not effective alone to treat fungal co-infections in COVID-19 patients managed in intensive care unit with prolonged immunomodulatory treatments [27,28] and have caused important side effects [29,30]. Triazoles cause hepatotoxicity, drug-drug interactions, QTc prolongation (the heart muscle takes a comparatively longer time to contract and relax than usual), skeletal fluorosis, pseudohyperaldosteronism, adrenal insufficiency, hyponatraemia and hypogonadism [29,30].”

  1. Nowadays, these kind of studies should be combined with others targeted to find out some clues about the mechanism of action of the new compound. Although natural products are a unvaluable source of new drugs, without more toxicity studies and explanations about their mechanism of action, these new compounds will be restricted to the lab.

We totally agree with the referee's statement, but there are several published articles that present promising results such as those obtained in our work as a starting point for other investigations. We can mention as examples the following articles: Saracino, I.M.; Foschi, C.; Pavoni, M.; Spigarelli, R.; Valerii, M.C.; Spisni, E. Antifungal Activity of Natural Compounds vs. Candida Spp.: A Mixture of Cinnamaldehyde and Eugenol Shows Promising In Vitro Results. Antibiotics 2022, 11, doi:10.3390/antibiotics11010073; Quirino, A.; Giorgi, V.; Palma, E.; Marascio, N.; Morelli, P.; Maletta, A.; Divenuto, F.; De Angelis, G.; Tancrè, V.; Nucera, S.; et al. Citrus Bergamia: Kinetics of Antimicrobial Activity on Clinical Isolates. Antibiotics 2022, 11, 1–15, doi:10.3390/antibiotics11030361; Silva, B.L.R.; Simão, G.; Campos, C.D.L.; Monteiro, C.R.A.V.; Bueno, L.R.; Ortis, G.B.; Mendes, S.J.F.; Moreira, I.V.; Maria-Ferreira, D.; Sousa, E.M.; et al. In Silico and In Vitro Analysis of Sulforaphane Anti-Candida Activity. Antibiotics 2022, 11, 1–18, doi:10.3390/antibiotics11121842; Chosidow, S.; Fantin, B.; Mascary, J.; Chau, F.; Verdier, M.; Rocheteau, P.; Gu, F.; Cattoir, V.; Lastours, V. De Synergistic Activity of Pep16 , a Promising New Antibacterial Pseudopeptide against Multidrug-Resistant Organisms , in Combination with Colistin against Multidrug-Resistant Escherichia Coli , In Vitro and in a Murine Peritonitis Model. 2023, 1–11, https://doi.org/10.3390/antibiotics12010081; Liu, Y.-Y.; Qin, Z.-H.; Yue, H.-Y.; Bergen, P.J.; Deng, L.-M.; He, W.-Y.; Zeng, Z.; Peng, X.; Liu, J. Synergistic Effects of Capric Acid and Colistin Against. Antibiotics 2022,1-11 https://doi.org/10.3390/antibiotics12010036.

The aim of our study was to evaluate the combination of ent-hardwickiic acid with low concentrations of amphotericin B against Candida strains.

Knowing that amphotericin B has been recommended for the treatment of pulmonary candidiasis associated with COVID-19 infection, but its side effects associated with several complications of SARS-CoV-2 infection, such as kidney injury, dyspnea, hypoxia, makes its use unfeasible, it is important to reduce the effective dose of this antibiotic to reduce the risk of side effects. Drug combination strategies have this potential to reduce the risk of side effects without loss of therapeutic efficacy.

In our study, the combination between ent-hardwickiic acid and amphotericin B markedly reduced the MIC values when compared with those of drugs alone. The combination was effective using lower concentrations of each compound than those needed to achieve the same effect of each isolated compounds. The combination of 1 µg/ mL of amphotericin B with 6.25 µg/ mL of ent-hardwickiic acid killed all the cells of C. albicans resistant strain within four hours of incubation. This concentration of amphotericin B in plasma has not been associated with toxic effects and drug discontinuation. According to literature, the pharmacodynamic characteristics of amphotericin B indicate that after the administration of doses of 0.6 to 3.0 mg/kg of body weight/day of amphotericin B deoxycholate (Bristol-Myers Squibb), the mean maximum concentrations (Cmaxs) achieved in serum are 1.1 to 3.6 µg/ mL. In the presence of serum, amphotericin B loses its fungicidal activity, but remains with its fungistatic activity.

We have included in this revised version of this manuscript information on the mechanism of action of ent-hardwickiic acid against oral pathogenic bacteria in the discussion section (page 7) as follows: “With regard to ent-hardwickiic acid, there is only one study reporting the mechanism of action of this diterpene against Streptococcus mutans (ATCC 25175) and Porphyromonas gingivalis (ATCC 33277) [68]. The authors performed assays to determine cell membrane integrity by leakage through the bacterial membrane of nucleic acids and protein. The results indicated that the diterpene ent-hardwickiic acid damaged the S. mutans and P. gingivalis cell membranes, causing cellular component release followed by marked release of cytoplasmic material [68]. Further studied are necessary to elucidate the mechanism of antifungal action of this compound.”          

We take this opportunity to thank the editor and referees for the pertinent and very important observations made to improve the quality of this manuscript.

We have taken into consideration all the issues mentioned by the editor and referees and we thank you again for taking your precious time to improve our manuscript.

We thank you in advance for your consideration and we are looking forward to hearing from you soon.

Yours sincerely,

Prof. Dr. Niege Araçari Jacometti Cardoso Furtado

Round 2

Reviewer 2 Report

Dear authors

Thank you for your response to the questions.

Please answer the following:

-Table 2, must be C. albicans

-Point 15. Spectrophotometric techniques do not allow differentiating between living and dead cells. Please enter the absorbances and the time the test lasted. In addition, in the MIC alone they analyzed viability with resazurin; why was this same compound not used in the checkboard microdilution methods?

Reviewer 3 Report

The authors have addressed one of the issues:

The potential of the compound has been shown. And it is very promising.

  1. However, more toxicity studies must be done, not only in normal cells (the authors mention absence of toxicity on cancer cells) but in animals. SOLVED
  2. Besides, the most used antifungal drugs are azoles and the authors do not show any experiment in combination with azoles. Although the explanation is correct for the covid-19 treatments, it is still necessary for the treatment of other infections. 
  3. Nowadays, these kind of studies should be combined with others targeted to find out some clues about the mechanism of action of the new compound. Although natural products are a unvaluable source of new drugs, without more toxicity studies and explanations about their mechanism of action, these new compounds will be restricted to the lab. The authors have put examples of other manuscripts accepted in the same Journal that do not show any experiments about the possible mechanism of action of the compound and they have also cited a reference with a described effect on the plasmatic membrane of bacteria. Candida membrane is different from bacterial membranes, due to the absence of sterols in bacterial membranes. Thus, it is important to see if the same effect is present for Candida membranes. 
  4.  

Round 3

Reviewer 3 Report

My opinionion about the manuscript is still the same. More experiments are needed to improve the quality of the work.